# Fabrication of Three-Dimensional ZnO: Ga@ITO@Ag SERS-Active Substrate for Sensitive and Repeatable Detectability

**DOI:** 10.3390/nano13010163

**Published:** 2022-12-29

**Authors:** Tung-Hao Chang, Yun-Ting Liu, Yu-Cheng Chang, An-Ya Lo

**Affiliations:** 1Department of Radiation Oncology, Changhua Christian Hospital, Changhua 50006, Taiwan; 2Department of Radiological Technology, Yuanpei University, Hsinchu 30015, Taiwan; 3Department of Medical Imaging and Radiological Sciences, Central Taiwan University of Science and Technology, Taichung 40601, Taiwan; 4Department of Materials Science and Engineering, Feng Chia University, Taichung 407102, Taiwan; 5Department of Chemical and Materials Engineering, National Chin-Yi University of Technology, Taichung 411030, Taiwan

**Keywords:** ZnO: Ga nanotowers, chemical bath, ITO layer, Ag nanoparticles, ion sputtering, surface-enhanced Raman scattering, amoxicillin

## Abstract

Vertically aligned ZnO: Ga nanotowers can be directly synthesized on a glass substrate with a ZnO seed film via the chemical bath method. A novel heterostructure of ZnO: Ga@ITO@Ag nanotowers was subsequently deposited in the ITO layer and Ag nanoparticles via the facile two-step ion-sputtering processes on the ZnO: Ga nanotowers. The appropriate ion-sputtering times of the ITO layer and Ag nanoparticles can benefit the fabrication of ZnO: Ga@ITO@Ag nanotowers with higher surface-enhanced Raman scattering (SERS) enhancement in detecting rhodamine 6G (R6G) molecules. Compared with ZnO: Ga@Ag nanotowers, ZnO: Ga@ITO@Ag nanotowers exhibited a high SERS enhancement factor of 2.25 × 10^8^ and a lower detection limit (10^−14^ M) for detecting R6G molecules. In addition, the ITO layer used as an intermediate layer between ZnO: Ga nanotowers and Ag nanoparticles can improve SERS enhancement, sensitivity, uniformity, reusability, detection limit, and stability for detecting amoxicillin molecules. This phenomenon shall be ascribed to the ITO layer exhibiting a synergistic Raman enhancement effect through interfacial charge transfer for enhancing SERS activity. As a result, ZnO: Ga@ITO@Ag nanotowers can construct a three-dimensional SERS substrate for potential applications in environmentally friendly and cost-effective chemical or drug detection.

## 1. Introduction

Raman spectroscopy is an optical detection technique that provides information on the characteristic spectra of analytes [1,2]. It has a wide range of applications in chemistry, physics, medicine, and biology for providing a fingerprint of molecular vibrations [3,4]. However, a significant disadvantage of Raman spectroscopy is that it is difficult for ordinary Raman spectroscopy to provide discernible signals for trace analytes due to the weak Raman scattering signal, resulting in weak Raman signals in most cases [5,6]. Surface-enhanced Raman spectroscopy (SERS) compensates for this deficiency through the plasmonic resonance of metallic nanostructures [7,8]. Due to localized surface plasmon resonances, molecules adsorbed on the nanostructured metal surface transmit a huge amplification of the electromagnetic (EM) field and lead to an order-of-magnitude increase in Raman scattering efficiency, further greatly enhancing the Raman signal [9,10]. In addition, SERS enables ultrasensitive and highly specific label-free detection to achieve high SERS enhancement factors [11,12,13,14]. Therefore, researchers have devoted themselves to developing various metal (such as Ag or Au) nanostructures for enhancing the local EM field [15,16,17]. Further, metallic SERS structures, hybrid structures composed of metallic/semiconductor materials, are also employed as SERS sensors [7,18,19,20]. Therefore, metal/semiconductor materials have attracted the attention of many researchers, mainly using their synergistic effect to enhance Raman signals significantly [21]. Efficient interfacial charge transfer in semiconductor materials can generate greater molecular polarization for SERS activity [22]. Nevertheless, their SERS enhancement is insignificant, whereas noble metals can enhance the local electric field by exciting the localized surface plasmon resonance (LSPR). Therefore, combining these two components can significantly enhance the Raman signal [8,18].

Recently, combining two-dimensional (2D) materials with metal plasmonic nanostructures to form a hybrid SERS substrate has become an emerging research topic [23,24]. A SERS substrate with a layer of 2D material provides a synergistic Raman enhancement effect of 2D material and plasmon resonance [25,26]. Among many 2D materials, graphene is commonly used to enhance Raman signals [27,28]. However, the preparation of graphene often requires chemical vapor deposition to be beneficially deposited on the substrate, and its uniformity and layer number also need to be effectively controlled [29,30]. In contrast, indium tin oxide (ITO) thin films have become essential for the substrate due to their excellent optoelectronic properties [31,32]. Furthermore, ITO is an n-type semiconductor with a wide bandgap of 3.3 to 4.3 eV at room temperature and exhibits a higher carrier density [33]. Therefore, the combination of ITO and Ag can increase carrier density, which is beneficial to improve its SERS enhancement effect [34]. However, in past research, ITO and Ag nanoparticles were not deposited sequentially on ZnO nanostructures for SERS application.

In the present study, the two-step ion-sputtering processes can be used to deposit a thin ITO layer and Ag nanoparticles on the vertically aligned ZnO: Ga nanotowers to form three-dimensional (3D) ZnO: Ga@ITO@Ag nanotowers as a high-performance SERS substrate. As a result, this SERS substrate exhibited higher SERS enhancement, high sensitivity, low detection limit, high uniformity, high reusability, and high stability for detecting amoxicillin molecules. Thus, the ZnO: Ga@ITO@Ag nanotowers have potential for application in detecting different fields.

## 2. Materials and Methods

### 2.1. Preparation of ZnO Seed Film

Glass substrates (7.5 cm × 2.5 cm) were soaked in ethanol and cleaned with an ultrasonic vibrator for 10 min to remove particles and organic contaminants from the substrate surface. Next, a ZnO seed film was prepared by spin-coating a layer of 20 mM zinc acetate dihydrate (97%, Alfa Aesar, Haverhill, MA, USA) solution in ethanol, followed by thermal annealing this substrate at 80 °C for 3 min and 350 °C for 20 min, respectively. Finally, the glass substrate with a ZnO seed film was cut into 1 cm × 1 cm for growing ZnO: Ga nanotowers.

### 2.2. Preparation of ZnO: Ga@ITO@Ag Nanotowers

The ZnO: Ga nanotowers can be directly grown on the glass substrate with a ZnO seed film via the chemical bath method in 100 mL of an aqueous solution containing 10 mM zinc nitrate hexahydrate (98%, Alfa Aesar, USA), 10 mM hexamethylenetetramine (HMTA, 99%, Alfa Aesar, USA), 1 mL 1,3-diaminopropane (DAP, 98%, Alfa Aesar, USA), and 1.5 mM gallium (Ⅲ) nitrate hydrate (98%, Alfa Aesar, USA), respectively. The glass substrates with a ZnO seed film were pasted on a glass sheet, placed in a sealed crystallizing dish (150 mL) containing the above reaction solution, and heated in a hotplate at T = 90 °C for 3 h. ITO layer was deposited on the ZnO: Ga nanotowers by ion-sputtering technique with RF 50 W at a ∼3.8 × 10^−6^ Torr pressure for the different deposition times. Ag nanoparticles were deposited on the ZnO: Ga nanotowers or ZnO: Ga@ITO nanotowers by ion-sputtering technique with DC 20 W at a ∼3.8 × 10^−6^ Torr pressure for the different deposition times. The synthesis procedure of the ZnO: Ga@ITO@Ag nanotowers is illustrated in Figure 1.

### 2.3. Characteristics

The surface morphologies of as-synthesized samples were examined by field-emission scanning electron microscopy (FESEM, Hitachi S-4800, Tokyo, Japan), operating at a 15 kV accelerating voltage. The microstructures of as-synthesized samples were examined by field-emission transmission electron microscopy (FETEM, JEOL-2100F, Tokyo, Japan), operating at a 200 kV accelerating voltage. The crystallinity of the ZnO: Ga@ITO nanotowers was analyzed by X-ray diffraction (XRD) utilizing a Bruker D2 phaser X-ray diffractometer (USA). X-ray photoelectron spectroscopy (XPS, ULVAC-PHI PHI 5000 Versaprobe II system, Chigasaki, Japan) was used to identify the surface elemental composition and electron configuration of ZnO: Ga@ITO nanotowers prepared on the glass substrate with a ZnO seed film.

### 2.4. SERS Testing

A confocal Raman microscope (MRI532S, Protrustech, Taiwan) equipped with a He-Ne laser with an excitation wavelength of 532 nm was used to measure the Raman spectra. Rhodamine 6G (R6G) and amoxicillin molecules act as well-known chemical and antibiotic targets to evaluate the SERS performance of ZnO: Ga@ITO@Ag nanotowers. R6G and amoxicillin were dissolved in the mixed solution with deionized water and ethanol. The as-prepared SERS substrates were immersed in target solutions with different concentrations for 1 h at room temperature and then dried by air purge. Use the single-point measurement mode with an acquisition time of 0.15 s to scan 5 times and take the average value of 10 random positions on the substrate as the final result. The SERS enhancement factor (EF) of the SERS substrate is calculated by EF = (I_SERS_ × C_R_)/(I_R_ × C_SERS_), where I_SERS_ and I_R_ are the intensity of target peaks (R6G at 1648 cm^−1^ or amoxicillin at 1351 cm^−1^) obtained from the SERS and Raman spectra. The C_SERS_ and C_R_ are the analyte concentrations used to measure the SERS and Raman spectra [18,35].

## 3. Results and Discussion

Figure 2a shows a tilt-view FESEM image of the ZnO: Ga nanotowers grown by the chemical bath method on a glass substrate with a ZnO seed film. The average sizes and lengths of ZnO: Ga nanotowers are about 125 nm and 7.6 μm, respectively. Figure 2b–f show the tilt-view FESEM images of the ZnO: Ga nanotower-deposited Ag nanoparticles under different ion-sputtering times. The ion-sputtering times are 15, 30, 60, 90, and 120 s, respectively. With the elongation of the ion-sputtering time of Ag nanoparticles, the sizes of Ag nanoparticles tend to increase gradually. However, when the ion-sputtering times exceed 90 s, the Ag nanoparticles on the tips of the nanotowers form a thin film. This phenomenon may further reduce the hotspot effect of ZnO: Ga nanotowers, thereby reducing the SERS enhancement effect.

The excellent three-dimensional structures of ZnO: Ga nanotowers can be beneficial in depositing a uniform ITO film using the ion-sputtering method. However, the ITO film is too thin to be observed in FESEM, so high-resolution TEM (HRTEM) and Energy Dispersive X-ray Spectroscopy (EDS) elemental mapping images are carried out to confirm its existence. Figure 3a reveals the FETEM image of a ZnO: Ga nanotower decorated with an ITO layer at an ion-sputtering time of 1 min. Figure 3b shows the HRTEM image of a ZnO: Ga@ITO nanotower with single crystalline, demonstrated by observed d-spacing of 0.265 nm, which is very consistent with the lattice spacing in the hexagonal (002) plane of ZnO (JCPDS Card No. 36-1451). In addition, an amorphous ITO layer with a thickness of about 3 nm is decorated on the surface of the ZnO: Ga nanotower. The combination of the HRTEM image and selected-area electron diffraction (SAED) pattern (Figure 3c) of a ZnO: Ga@ITO nanotower indicates that the ZnO: Ga nanotower grew along the [001] direction. The element distributions of a ZnO: Ga@ITO nanotower were obtained by analyzing the EDS mapping images, as shown in Figure 3d. It can clearly depict the arrangement of an ITO thin film, while In and Sn are homogeneously distributed on a ZnO: Ga@ITO nanotower. In addition, the presence of the Ga element can also prove that Ga has been doped into the ZnO nanotower. Furthermore, the overall crystallinity of the ZnO: Ga@ITO nanotowers was analyzed by X-ray powder diffraction. The XRD pattern (Figure 3e) of ZnO: Ga@ITO nanotowers appeared with three diffraction peaks at 2θ values of 34.5°, 63.0°, and 72.7°, which can be indexed to (002), (103), and (004) crystal faces of the hexagonal-phase ZnO (JCPDS Card No. 36–1451), respectively.

The surface chemical composition and elemental valence states of ZnO: Ga@ITO nanotowers grown on the glass substrate with a ZnO seed film were further analyzed by XPS. Figure 4a shows that the survey scan spectrum can confirm the presence of Zn, O, Ga, In, and Sn elements. The presence of C 1 s shall be ascribed to the pump oil in the vacuum system of the XPS equipment or the organic layer coating the surface of ZnO: Ga@ITO nanotowers [36]. The high-resolution In 3d spectrum (Figure 4b) shows that two peaks at 443.8 and 451.4 eV correspond to the In 3d_5/2_ and 3d_3/2_, respectively, with binding energies of In(III) [37]. The high-resolution Sn 3d spectrum (Figure 4c) reveals that two peaks at 485.6 and 494.2 eV correspond to the Sn 3d_5/2_ and 3d_3/2_, respectively, with binding energies of Sn(IV) [37].

Figure 5a reveals an FESEM image of ZnO: Ga@ITO nanotowers decorated with Ag nanoparticles at an ion-sputtering time of 60 s. The Ag nanoparticles were coated entirely on the surface of ZnO: Ga@ITO nanotowers. The average sizes of Ag nanoparticles are about 5–35 nm. Figure 5b shows an FETEM image of a ZnO: Ga@ITO@Ag nanotower. Compared with the FESEM image, the Ag nanoparticles on the FETEM image are significantly reduced. This phenomenon is attributed to shaking off by ultrasonic vibration during the preparation of the TEM specimen. The HRTEM image (Figure 5c) reveals that the lattice fringe with a spacing of 0.236 nm corresponds to the (111) lattice plane of cubic-phase Ag (JCPDS Card No. 89-0722). In addition, the EDS mapping images (Figure 5d) of a ZnO: Ga@ITO@Ag nanotower can confirm the heterostructured construction by Zn, O, Ga, In, Sn, and Ag. The EDS spectrum (Figure 5e) of a ZnO: Ga@ITO@Ag nanotower with Zn, O, Ga, In, Sn, Ag, and Cu (Cu arises from TEM grid) can be used to evaluate the elemental analysis. This result indicates no other impurity elements in the ZnO: Ga@ITO@Ag nanotower.

The SERS activity is susceptible to the morphology and density of metal nanoparticles. Therefore, ZnO: Ga nanotowers decorated Ag nanoparticles with different ion-sputtering times as substrates to detect the SERS activity. Rhodamine 6G (R6G) was used as a probe molecule due to its recognized vibrational characteristics. Figure 6a shows the SERS spectra of ZnO: Ga nanotowers with different Ag ion-sputtering times that were immersed in 1 × 10^−6^ M R6G solution for 1 h and then dried with air purge. The main vibrational mode for the characteristic peaks of the R6G molecule is C−H in-plane bending (1127 and 1186 cm^−1^), C−O−C in-plane bending (1311 cm^−1^), and aromatic C−C stretch (1362, 1421, 1510, 1572, and 1649 cm^−1^), respectively [38,39]. The SERS activities gradually improved with the ion-sputtering time, ranging from 15 to 60 s, but decreased when further increasing the ion-sputtering time to 90 and 120 s. Therefore, it can be obtained that the strongest SERS activity is at an ion-sputtering time of 60 s.

Figure 6b shows that the SERS spectra of ZnO: Ga@ITO@Ag nanotowers with different ITO ion-sputtering times at Ag ion-sputtering time of 60 s were immersed in 1 × 10^−6^ M R6G solution for 1 h and then dried with air purge. The SERS activities gradually improved with the ion-sputtering time ranging from 0.5 to 1 min but decreased with further increasing the ion-sputtering time to 3 and 5 min. Therefore, it can be obtained that the strongest SERS activity is at an ion-sputtering time of 1 min. This reason is attributed to the better distribution of hotspots provided by a sputtering time of 1 min, thus, significantly enhancing its SERS activity. In order to verify that the ITO layer as the intermediate layer between ZnO: Ga nanotowers and Ag nanoparticles has better SERS activity than other materials, herein, we selected three different materials (such as ZnO/Ga_2_O_3_, TiO_2_, and MgO) as the intermediate layer between ZnO: Ga nanotowers and Ag nanoparticles to evaluate their SERS activities at the same ion-sputtering time of 1 min. Figure 6c shows that the SERS spectra of as-synthesized substrates with different materials as the intermediate layer were immersed in 1 × 10^−6^ M R6G solution for 1 h and then dried with air purge. The results show that the intermediate layer of ITO can exhibit the best SERS activity. This phenomenon is ascribed to the ITO layer exhibiting a synergistic Raman enhancement effect by interfacial charge transfer for enhancing SERS activity [25,26].

To further evaluate the limit of detection (LOD) of ZnO: Ga@Ag nanotowers and ZnO: Ga@ITO@Ag nanotowers, the SERS spectra for the as-synthesized SERS substrates were immersed in different concentrations of R6G solution for 1 h and then dried with air purge. Figure 7a shows the SERS spectra of ZnO: Ga@Ag nanotowers at a concentration of R6G solution from 10^−7^ to 10^−11^ M. The LOD of ZnO: Ga@Ag nanotowers reached 10^−11^ M. In addition, the SERS spectra (Figure 7b) of ZnO: Ga@ITO@Ag nanotowers were at an R6G solution concentration from 10^−10^ to 10^−14^ M. The LOD of ZnO: Ga@Ag nanotowers reached 10^−14^ M. The LOD of ZnO: Ga@ITO@Ag nanotowers can be remarkably improved by 3 orders compared with ZnO: Ga@Ag nanotowers, indicating that this substrate has a high sensitivity to detect R6G molecule. The SERS enhancement factor (EF) values of 1649 cm^−1^ for ZnO: Ga@Ag nanotowers and ZnO: Ga@ITO@Ag nanotowers were calculated to be 1.57 × 10^8^ and 2.25 × 10^8^, respectively [40,41,42]. The SERS EF of ZnO: Ga@ITO@Ag nanotowers is 1.43-times higher than the ZnO: Ga@Ag nanotowers. Previous studies have shown that the metal–semiconductor hybrid structure prevents the recombination of electron–hole pairs, which significantly improves SERS performance [20,43]. Compared with ZnO: Ga@Ag nanotowers, ZnO: Ga@ITO@Ag nanotowers have an ITO layer that is more conducive to transferring electron–hole pairs so that the SERS performance can be further improved.

The uniformity of the SERS substrate plays a significant role in its practical application. Herein, the SERS spectra (Figure 8a,b) of 10 random points were measured on the ZnO: Ga@Ag nanotowers and ZnO: Ga@ITO@Ag nanotowers with immersed (10^−6^ M) R6G solution for 1 h, respectively. As a result, the peak positions and intensities of the R6G Raman signals are very similar at different positions of ZnO: Ga@Ag nanotowers and ZnO: Ga@ITO@Ag nanotowers. In order to confirm that the as-synthesized SERS substrates (ZnO: Ga@Ag nanotowers and ZnO: Ga@ITO@Ag nanotowers) can also be used to detect antibiotics with good uniformity, Amoxicillin was selected, as it is an antibiotic molecule widely used to treat bacterial infections in humans and animals [8]. Herein, the SERS spectra (Figure 8c,d) of ten random points were measured on the ZnO: Ga@Ag nanotowers and ZnO: Ga@ITO@Ag nanotowers with immersed (10^−3^ and 10^−6^ M) amoxicillin solution for 1 h, respectively. The main vibrational mode for the characteristic peaks of the amoxicillin molecule is benzene-ring deformation (668 cm^−1^), plane deformation (790 cm^−1^), benzene-ring breathing (865 cm^−1^), amine bending (935 cm^−1^), C−H stretching (1038 cm^−1^), benzene-plane deformation (1288 cm^−1^), amine twisting (1351 cm^−1^), CH_3_ asymmetric bending (1490 cm^−1^), C−C stretching (1603 cm^−1^), and N−H bending (1651 cm^−1^), respectively [18]. In addition, the peak positions and intensities of the amoxicillin Raman signals are also similar at different positions of ZnO: Ga@Ag nanotowers and ZnO: Ga@ITO@Ag nanotowers. This result can also prove that the as-synthesized three-dimensional SERS substrates exhibit a high uniformity for detecting R6G or amoxicillin. The SERS EF values of 1351 cm^−1^ for ZnO: Ga@Ag nanotowers and ZnO: Ga@ITO@Ag nanotowers were calculated to be 1.25 × 10^5^ and 2.33 × 10^8^, respectively. The SERS EF of ZnO: Ga@ITO@Ag nanotowers is 1864-times higher than the ZnO: Ga@Ag nanotowers. This result shows that the deposition of the ITO layer is more conducive to transferring electron–hole pairs so that the SERS detection of amoxicillin molecules can further improve.

Raman signal reproducibility is also an important indicator for evaluating SERS substrates. Figure 9a,b show the results of the reversible SERS behavior of the ZnO: Ga@Ag nanotowers and ZnO: Ga@ITO@Ag nanotowers over five cycles for detecting R6G (10^−6^ M) solution. After UV light (253.7 nm, 36 W) irradiation, R6G was detected from the ZnO: Ga@Ag nanotowers and ZnO: Ga@ITO@Ag nanotowers. Due to the photodegradation of products with small molecules, they can be removed from the SERS substrate by washing, and the SERS substrate is not contaminated. The two SERS substrates maintained excellent activity after five degradation cycles for the R6G solution. Figure 9c shows the result of the reversible SERS behavior of the ZnO: Ga@Ag nanotowers over five cycles for detecting amoxicillin (10^−3^ M) solution. The Raman signal of amoxicillin cannot be detected after one cycle. Figure 9d reveals the result of the reversible SERS behavior of the ZnO: Ga@ITO@Ag nanotowers over five cycles for detecting amoxicillin (10^−5^ M) solution. ZnO: Ga@ITO@Ag nanotowers still maintain excellent activity after five degradation cycles. This result reveals that the ZnO: Ga@ITO@Ag nanotowers are highly reproducible and still exhibit a similar intensity of Raman signal after five cycles. This phenomenon can also prove that the ion-sputtering ITO layer can improve the SERS enhancement and reusability for detecting amoxicillin molecules.

The SERS spectra for the as-synthesized SERS substrates were immersed in different concentrations of amoxicillin solution for 1 h and then dried with air purge. It evaluated the LOD of ZnO: Ga@Ag nanotowers and ZnO: Ga@ITO@Ag nanotowers for amoxicillin solution. Figure 10a shows the SERS intensities of ZnO: Ga@Ag nanotowers and ZnO: Ga@ITO@Ag nanotowers at 1351 cm^−1^ in a concentration range of 10^−6^–10^−10^ M of amoxicillin solution. With a decrease in amoxicillin concentration, the SERS signal intensities at 1351 cm^−1^ gradually decreased. When the concentration of amoxicillin solution is lower than 10^−6^ M, the SERS signal at 1351 cm^−1^ cannot be effectively detected for ZnO: Ga@Ag nanotowers. However, the SERS signal at 1351 cm^−1^ can still be detected, even at a concentration of amoxicillin of 10^−10^ M for ZnO: Ga@ITO@Ag ZnO nanotowers. This result proves that ITO is the intermediate layer between ZnO: Ga nanotowers and Ag nanoparticles, which can provide LOD for detecting R6G (10^−14^ M) or amoxicillin (10^−10^ M) solution. In order to evaluate the temporal stability of ZnO: Ga@Ag nanotowers and ZnO: Ga@ITO@Ag nanotowers, the amoxicillin (10^−3^ and 10^−6^ M) solution was detected after preparing SERS substrates and stored in the dark for 1, 5, 10, 15, and 20 days. Figure 10b shows the SERS intensities of ZnO: Ga@Ag nanotowers and ZnO: Ga@ITO@Ag nanotowers at 1351 cm^−1^ for different detection times. After ten days, the intensities of the SERS signals at 351 cm^−1^ remained at 71.5% (ZnO: Ga@ITO@Ag nanotowers) and 38.3% (ZnO: Ga@Ag nanotowers), respectively. ZnO: Ga@Ag nanotowers did not exhibit any SERS signal at 1351 cm^−1^ after 20 days. For ZnO: Ga@ITO@Ag nanotowers, the intensities of the SERS signals at 351 cm^−1^ remained at 26.3%. This result proves that ITO is the intermediate layer between ZnO: Ga nanotowers and Ag nanoparticles, which can also improve their stability for detecting amoxicillin molecules.

## 4. Conclusions

In this study, we successfully fabricated a high-performance 3D SERS substrate on glass substrates with a ZnO seed film composed of ZnO: Ga nanotower cores, intermediate ITO layer, and Ag nanoparticle sheaths (ZnO: Ga@ITO@Ag nanotowers) by combining the chemical bath and ion-sputtering methods. Furthermore, the appropriate ion-sputtering times of the ITO layer and Ag nanoparticles were optimized to produce significant SERS enhancement in the R6G molecules. As a result, ZnO: Ga@ITO@Ag nanotowers exhibited a higher SERS EF (2.25 × 10^8^) and a lower detection limit (10^−14^ M) than ZnO: Ga@Ag nanotowers for detecting R6G molecules. Furthermore, the ITO layer used as an intermediate layer between ZnO: Ga nanotowers and Ag nanoparticles is critical in improving SERS enhancement, sensitivity, uniformity, reusability, detection limit, and stability for detecting amoxicillin molecules. ZnO: Ga@ITO@Ag nanotowers exhibited some advantages of a facile preparation process, high sensitivity, low detection limit, high reusability, high uniformity, and high stability, which are conducive to the actual application of SERS in different fields.

## Figures and Tables

**Figure 1 nanomaterials-13-00163-f001:**
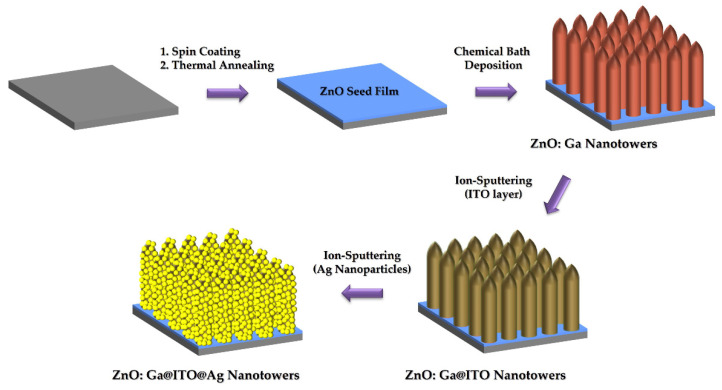
Schematic representation of the synthesis procedure of ZnO: Ga@ITO@Ag nanotowers.

**Figure 2 nanomaterials-13-00163-f002:**
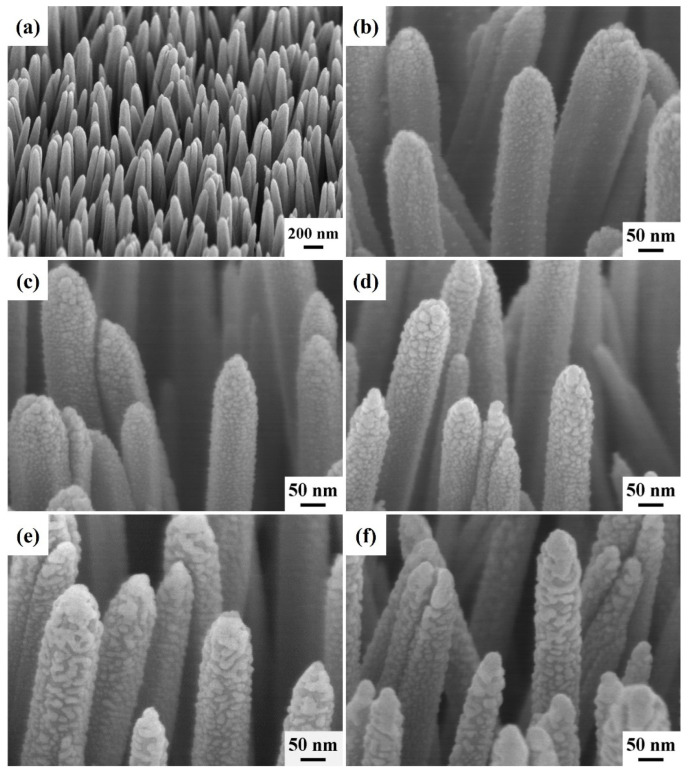
Tilt-view FESEM images of (**a**) ZnO: Ga nanotowers and (**b**–**f**) ZnO: Ga@Ag nanotowers with different Ag ion-sputtering times grown on glass substrates with a ZnO seed film. The ion-sputtering times are (**b**) 15, (**c**) 30, (**d**) 60, (**e**) 90, and (**f**) 120 s, respectively.

**Figure 3 nanomaterials-13-00163-f003:**
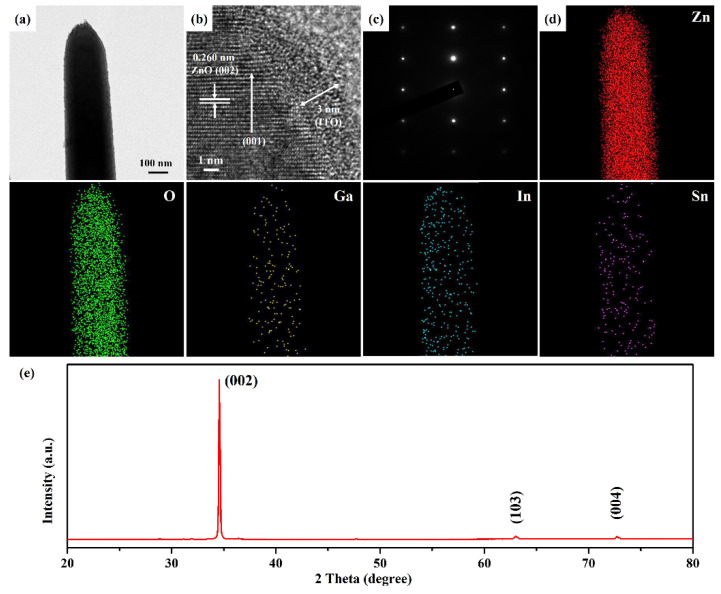
(**a**) FETEM, (**b**) HRTEM, (**c**) SAED pattern, and (**d**) EDS mapping images of a ZnO: Ga@ITO nanotower. (**e**) The XRD pattern of ZnO: Ga@ITO nanotowers grown on glass substrate with a ZnO seed film.

**Figure 4 nanomaterials-13-00163-f004:**
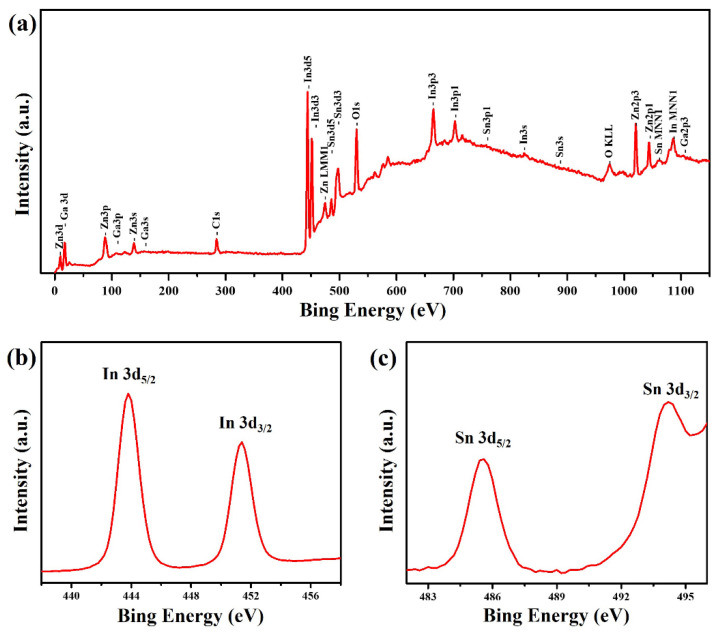
XPS spectra of the ZnO: Ga@ITO nanotowers: (**a**) survey spectrum, (**b**) In 3d, and (**c**) Sn 3d.

**Figure 5 nanomaterials-13-00163-f005:**
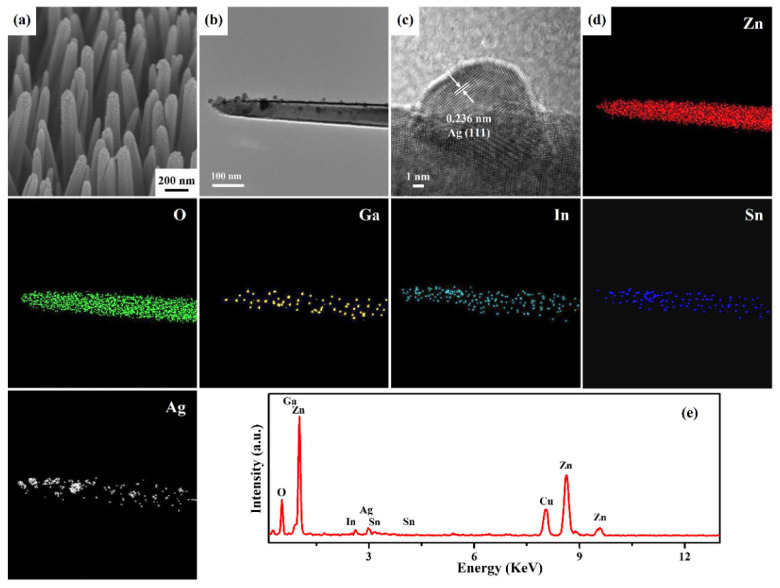
(**a**) The tilt-view FESEM image of ZnO: Ga@ITO@Ag nanotowers. (**b**) FETEM, (**c**) HRTEM, (**d**) EDS mapping images, and (**e**) EDS spectrum of a ZnO: Ga@ITO@Ag nanotower.

**Figure 6 nanomaterials-13-00163-f006:**
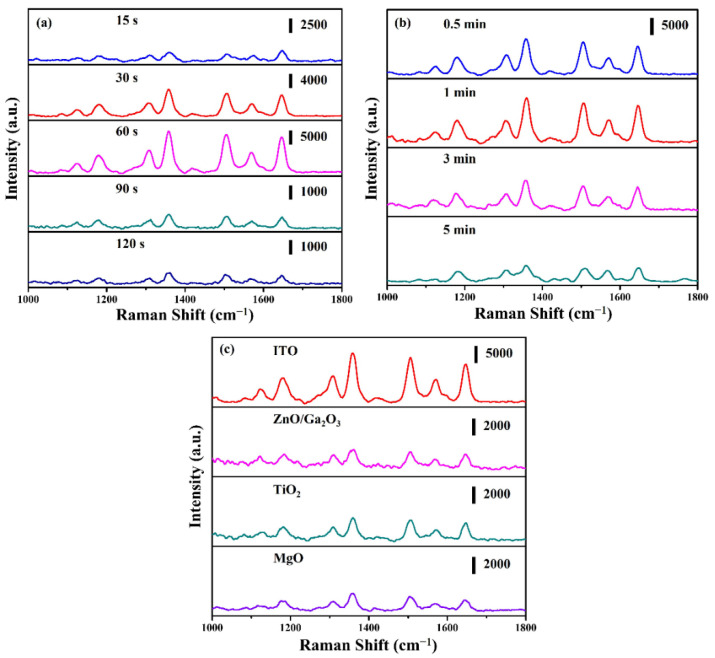
(**a**) SERS spectra of (10^−6^ M) R6G solution on ZnO: Ga@Ag nanotowers fabricated at different Ag ion-sputtering times. (**b**) SERS spectra of (10^−6^ M) R6G solution on ZnO: Ga@ITO@Ag nanotowers fabricated at different ITO ion-sputtering times. (**c**) SERS spectra of (10^−6^ M) R6G solution on the different materials for the intermediate layer between ZnO: Ga nanotowers and Ag nanoparticles.

**Figure 7 nanomaterials-13-00163-f007:**
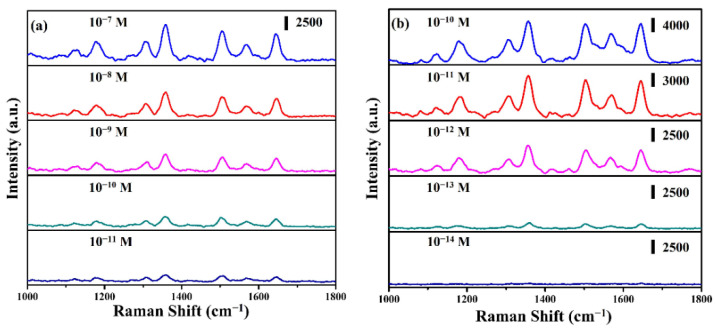
SERS spectra of R6G solution at different concentrations were obtained from (**a**) ZnO: Ga@Ag nanotowers and (**b**) ZnO: Ga@ITO@Ag nanotowers.

**Figure 8 nanomaterials-13-00163-f008:**
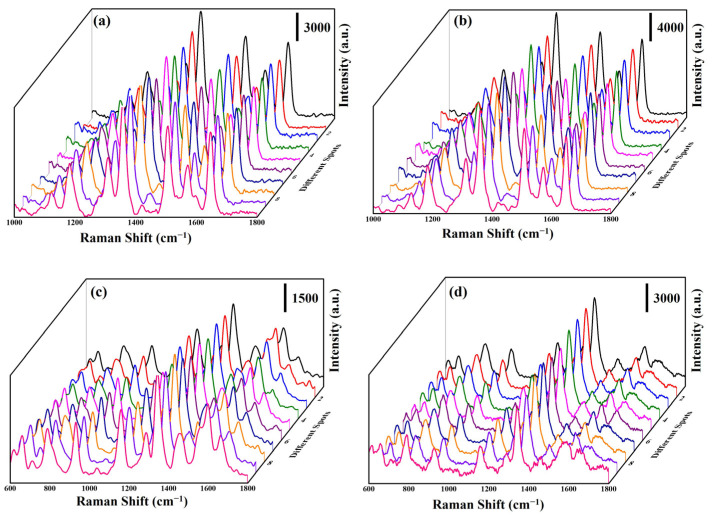
SERS spectra of (10^−6^ M) R6G solution at 10 random points on (**a**) ZnO: Ga@Ag nanotowers and (**b**) ZnO: Ga@ITO@Ag nanotowers. SERS spectra of (10^−3^ and 10^−6^ M) amoxicillin solution at 10 random points on (**c**) ZnO: Ga@Ag nanotowers and (**d**) ZnO: Ga@ITO@Ag nanotowers, respectively.

**Figure 9 nanomaterials-13-00163-f009:**
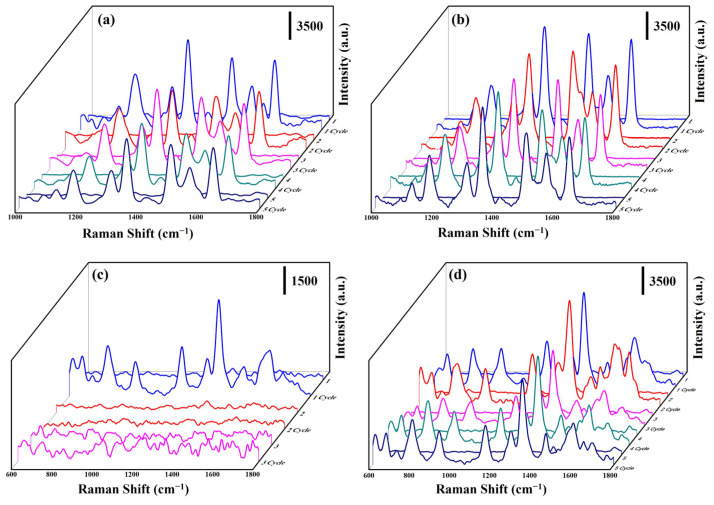
SERS spectra of (10^−6^ M) R6G solution on (**a**) ZnO: Ga@Ag nanotowers and (**b**) ZnO: Ga@ITO@Ag nanotowers at the five cycles. SERS spectra of (10^−3^ and 10^−6^ M) amoxicillin solution on (**c**) ZnO: Ga@Ag nanotowers and (**d**) ZnO: Ga@ITO@Ag nanotowers at the five cycles.

**Figure 10 nanomaterials-13-00163-f010:**
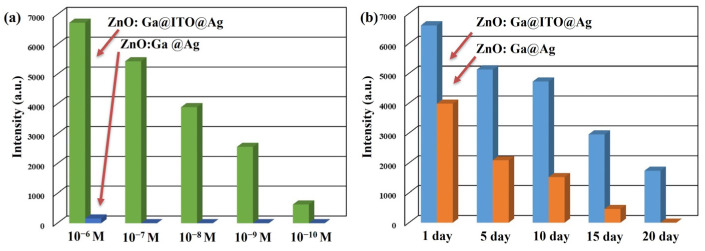
(**a**) The SERS intensities of ZnO: Ga@Ag nanotowers and ZnO: Ga@ITO@Ag nanotowers at 1351 cm^−1^ under the different concentrations of amoxicillin solution. (**b**) The SERS intensities of ZnO: Ga@Ag nanotowers and ZnO: Ga@ITO@Ag nanotowers at 1351 cm^−1^ under different detection times.

## Data Availability

Not applicable.

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
