# Peer review of "Fabrication of Three-Dimensional ZnO: Ga@ITO@Ag SERS-Active Substrate for Sensitive and Repeatable Detectability"

_nanomaterials, 2022, doi:10.3390/nano13010163_

Round 1

Reviewer 1 Report

In the current work, Tung-Hao Chang et al. used a ZnO seed film, a chemical bath technique, and an unique ZnO:Ga heterostructure to directly synthesis the vertically aligned ZnO:Ga nanotowers on the glass substrate. On the ZnO: Ga nanotowers, an easy two-step ion-sputtering technique was used to deposit Ga@ITO@Ag nanotowers in the ITO layer and Ag nanoparticles. To construct ZnO: Ga@ITO@Ag nanotowers with better surface-enhanced Raman scattering (SERS) enhancement for detecting rhodamine 6G (R6G) molecules, the ITO layer and Ag nanoparticles may benefit from the proper ion-sputtering periods. After a small adjustment, the current work is nicely written and ready for publication.

1.       In introduction, The author needs to update the following reference and emphasize the significance of heterostructure in the SERS application. (Sensors and Actuators B 207, 430-436 (2015))

2.       Details of the SERS EF calculation should be included to the experimental section.

3.       The number of spectra measurements for each sample should be added by the author.

4.       Importantly, the discussion part should include the SERS mechanism of heterostructure.

Author Response

In the current work, Tung-Hao Chang et al. used a ZnO seed film, a chemical bath technique, and an unique ZnO:Ga heterostructure to directly synthesis the vertically aligned ZnO:Ga nanotowers on the glass substrate. On the ZnO: Ga nanotowers, an easy two-step ion-sputtering technique was used to deposit Ga@ITO@Ag nanotowers in the ITO layer and Ag nanoparticles. To construct ZnO: Ga@ITO@Ag nanotowers with better surface-enhanced Raman scattering (SERS) enhancement for detecting rhodamine 6G (R6G) molecules, the ITO layer and Ag nanoparticles may benefit from the proper ion-sputtering periods. After a small adjustment, the current work is nicely written and ready for publication.

Response: Thanks for the pertinent and positive comments.

  1. In introduction, The author needs to update the following reference and emphasize the significance of heterostructure in the SERS application. (Sensors and Actuators B 207, 430-436 (2015))

Response: Thanks for your reminder. We have updated this reference in the revised manuscript.

  1. Details of the SERS EF calculation should be included to the experimental section.

Response: Thanks for your reminder. We have added this description to the revised manuscript. The SERS enhancement factor (EF) of the SERS substrate is calculated by EF = (ISERS × CR)/(IR × CSERS), where ISERS and IR are the intensity of target peaks (R6G at 1648 cm−1 or amoxicillin at 1351 cm−1) obtained from the SERS and Raman spectra. The CSERS and CR are the analyte concentrations used to measure the SERS and Raman spectra [1,2].

  1. The number of spectra measurements for each sample should be added by the author.

Response: Thanks for your reminder. We have amended this description in the revised manuscript. Use the single-point measurement mode with an acquisition time of 0.15 s to scan 5 times and take the average value of 10 random positions on the substrate as the final result.

  1. Importantly, the discussion part should include the SERS mechanism of heterostructure.

Response: Thanks for your reminder. Previous studies have shown that the metal-semiconductor hybrid structure prevents the recombination of electron-hole pairs, which significantly improves SERS performance [3,4]. Compared with ZnO: Ga@Ag nanotowers, ZnO: Ga@ITO@Ag nanotowers have an ITO layer that is more conducive to transferring electron-hole pairs so that the SERS performance can be further improved.

Reference

  1. Chang, T.-H.; Chuang, K.-W.; Chang, Y.-C. Ag/Ga-doped ZnO/pyramidal silicon as a multifunctional surface-enhanced Raman scattering substrate. Journal of Alloys and Compounds 2022, 893, 162288, doi:https://doi.org/10.1016/j.jallcom.2021.162288.
  2. Mercadal, P.A.; Encina, E.R.; Villa, J.E.L.; Coronado, E.A. A New Figure of Merit to Assess the SERS Enhancement Factor of Colloidal Gold Nanoparticle Aggregates. J. Phys. Chem. C 2021, 125, 4056-4065, doi:10.1021/acs.jpcc.0c09122.
  3. Sivashanmugan, K.; Liao, J.-D.; Liu, B.H.; Yao, C.-K.; Luo, S.-C. Ag nanoclusters on ZnO nanodome array as hybrid SERS-active substrate for trace detection of malachite green. Sensors and Actuators B: Chemical 2015, 207, 430-436, doi:https://doi.org/10.1016/j.snb.2014.10.088.
  4. Li, S.; Zhang, N.; Zhang, N.; Lin, D.; Hu, X.; Yang, X. Three-dimensional ordered Ag/ZnO/Si hierarchical nanoflower arrays for spatially uniform and ultrasensitive SERS detection. Sensors and Actuators B: Chemical 2020, 321, 128519, doi:https://doi.org/10.1016/j.snb.2020.128519.

Reviewer 2 Report

The article "Fabrication of Three-Dimensional ZnO: Ga@ITO@Ag SERS-Active Substrate for Sensitive and Repeatable Detectability" by Tung-Hao Chang et al. describes synthesis of novel SERS substrate. The overall scientific design looks valid, but the article still requires major revision:

1. The introduction section is very short for this relatively hot topic. The synthesis of hybrid metal-semiconductor (or nanomaterial) substrate require more description of state-of-the-art system and performance limits reachable by it.

2. Materials section require more description of Raman system used in this experiment

3. the deposition of the ITO layer is more conducive to the SERS detection of amoxicillin molecules - should be rephrased

Author Response

Reviewers' comments:

The article "Fabrication of Three-Dimensional ZnO: Ga@ITO@Ag SERS-Active Substrate for Sensitive and Repeatable Detectability" by Tung-Hao Chang et al. describes synthesis of novel SERS substrate. The overall scientific design looks valid, but the article still requires major revision:

Response: Thanks for the pertinent and positive comments.

  1. The introduction section is very short for this relatively hot topic. The synthesis of hybrid metal-semiconductor (or nanomaterial) substrate require more description of state-of-the-art system and performance limits reachable by it.

Response: Thanks for your reminder. We have amended this description in the revised manuscript. Besides metallic SERS structures, hybrid structures composed of metallic/semiconductor materials are also employed as SERS sensors [1-4]. Therefore, metal/semiconductor materials have attracted the attention of many researchers, mainly using their synergistic effect to enhance Raman signals significantly [5]. Efficient interfacial charge transfer in semiconductor materials can generate greater molecular polarization for SERS activity [6]. Nevertheless, their SERS enhancement is insignificant, whereas noble metals can enhance the local electric field by exciting the localized surface plasmon resonance (LSPR). Therefore, combining these two components can significantly enhance the Raman signal [1,7].

  1. Materials section require more description of Raman system used in this experiment

Response: Thanks for your reminder. We have added this description in the revised manuscript. A confocal Raman microscope (MRI532S, Protrustech, Taiwan) equipped with a He-Ne laser with an excitation wavelength of 532 nm was used to measure the Raman spectra. Rhodamine 6G (R6G) and amoxicillin molecules act as well-known chemical and antibiotic targets to evaluate the SERS performance of ZnO: Ga@ITO@Ag nanotowers. R6G and amoxicillin were dissolved in the mixed solution with deionized water and ethanol. The as-prepared SERS substrates were immersed in target solutions with different concentrations for 1 h at room temperature and then dried by air purge. Use the single-point measurement mode with an acquisition time of 0.15 s to scan 5 times and take the average value of 10 random positions on the substrate as the final result. The SERS enhancement factor (EF) of the SERS substrate is calculated by EF = (ISERS × CR)/(IR × CSERS), where ISERS and IR are the intensity of target peaks (R6G at 1648 cm−1 or amoxicillin at 1351 cm−1) obtained from the SERS and Raman spectra. The CSERS and CR are the analyte concentrations used to measure the SERS and Raman spectra [1,8].

  1. the deposition of the ITO layer is more conducive to the SERS detection of amoxicillin molecules - should be rephrased

Response: Thanks for your reminder. We have amended this description in the revised manuscript. This result shows that the deposition of the ITO layer is more conducive to transferring electron-hole pairs so that the SERS detection of amoxicillin molecules can further improve.

Reference:

  1. Chang, T.-H.; Chuang, K.-W.; Chang, Y.-C. Ag/Ga-doped ZnO/pyramidal silicon as a multifunctional surface-enhanced Raman scattering substrate. Journal of Alloys and Compounds 2022, 893, 162288, doi:https://doi.org/10.1016/j.jallcom.2021.162288.
  2. Chang, T.-H.; Chuang, K.-W.; Chang, Y.-C.; Chou, C.-M. Optimizing and improving the growth of Ag nanoparticles decorated on the silicon pyramid for surface-enhanced Raman spectroscopy. Materials Chemistry and Physics 2022, 280, 125823, doi:https://doi.org/10.1016/j.matchemphys.2022.125823.
  3. Chang, T.-H.; Chang, Y.-C.; Wu, S.-H. Ag nanoparticles decorated ZnO: Al nanoneedles as a high-performance surface-enhanced Raman scattering substrate. Journal of Alloys and Compounds 2020, 843, 156044, doi:https://doi.org/10.1016/j.jallcom.2020.156044.
  4. Sivashanmugan, K.; Liao, J.-D.; Liu, B.H.; Yao, C.-K.; Luo, S.-C. Ag nanoclusters on ZnO nanodome array as hybrid SERS-active substrate for trace detection of malachite green. Sensors and Actuators B: Chemical 2015, 207, 430-436, doi:https://doi.org/10.1016/j.snb.2014.10.088.
  5. Li, X.; Zhou, H.; Wang, L.; Wang, H.; Adili, A.; Li, J.; Zhang, J. SERS paper sensor based on three-dimensional ZnO@Ag nanoflowers assembling on polyester fiber membrane for rapid detection of florfenicol residues in chicken. Journal of Food Composition and Analysis 2023, 115, 104911, doi:https://doi.org/10.1016/j.jfca.2022.104911.
  6. Liu, Y.; Li, R.; Zhou, N.; Li, M.; Huang, C.; Mao, H. Recyclable 3D SERS devices based on ZnO nanorod-grafted nanowire forests for biochemical sensing. Applied Surface Science 2022, 582, 152336, doi:https://doi.org/10.1016/j.apsusc.2021.152336.
  7. Chang, Y.-C.; Wu, S.-H. Bi-functional Al-doped ZnO@SnO2 heteronanowires as efficient substrates for improving photocatalytic and SERS performance. Journal of Industrial and Engineering Chemistry 2019, 76, 333-343, doi:https://doi.org/10.1016/j.jiec.2019.03.058.
  8. Mercadal, P.A.; Encina, E.R.; Villa, J.E.L.; Coronado, E.A. A New Figure of Merit to Assess the SERS Enhancement Factor of Colloidal Gold Nanoparticle Aggregates. J. Phys. Chem. C 2021, 125, 4056-4065, doi:10.1021/acs.jpcc.0c09122.

Round 2

Reviewer 2 Report

All my former questions were answered and the article can be accepted after the minor English language/style corrections.